# Involvement of NF-κB in the reversal of CYP3A down-regulation induced by sea buckthorn in BCG-induced rats

**Fengting Liu**[☯]**, Tao Wang**[☯]**, Xiaoxia Li, Jinxue Jia, Qin Lin, Yongzhi Xue**[iD]*

Department of Pharmacology, Institute of Pharmacokinetics and Liver Molecular Pharmacology, Baotou Medical College, Baotou, Inner mongolia, China

☯ These authors contributed equally to this work.

* xyzhxyzh68@sohu.com

**Data Availability Statement:** All relevant data are within the manuscript and its Supporting Information files.

**Funding:** This research is supported by the following scientific research projects. In addition,

## Abstract

Previous studies reported that sea buckthorn (*Hippophae rhamnoides* L., Elaeagnaceae, HRP) exhibits hepatoprotective effects via its anti-inflammatory and antioxidant properties as well as its inhibitory effects on collagen synthesis. However, it is unclear whether this hepatoprotective effect is also achieved by regulating liver drug metabolism enzyme pathways. Herein, we examined the regulatory effect of HRP on cytochrome P450 3A (CYP3A) in rats with immune liver injury, and explored the molecular mechanism of its hepatoprotective effect. Rat models of immunological liver injury were induced by intravenous injections of *Bacillus Calmette-Guerin* (*BCG;* 125 mg kg$^{-1}$; 2 wks). Specific protein levels were detected by ELISA or western blot, and CYP3A mRNA expression was detected by RT-PCR. High-performance liquid chromatography (HPLC) detected relative changes in CYP3A metabolic activity based on the rates of 1-hydroxylation of the probe drug midazolam (MDZ). BCG pretreatment (125 mg kg-1) significantly down-regulated liver CYP3A protein expression compared with the control, metabolic activity, and transcription levels while up-regulating liver NF-κB, IL-1β, TNF-α and iNOS. HRP intervention (ED$_{50}$: 78 mg kg$^{-1}$) moderately reversed NF-κB, inflammatory cytokines, and iNOS activation in a dose-dependent manner ($P < 0.05$), and suppressed CYP3A down-regulation ($P < 0.05$); thereby partially alleviating liver injury. During immune liver injury, HRP may reverse CYP3A down-regulation by inhibiting NF-κB signal transduction, and protect liver function, which involves regulation of enzymes transcriptionally, translationally and post-translationally. The discovery that NF-κB is a molecular target of HRP may initiate the development and optimization of a clinical therapeutic approach to mitigate hepatitis B and other immunity-related liver diseases.

## Introduction

Nearly all viral hepatitis are caused by the hepatitis A–E viruses [1]. Previous studies have focused primarily on the development of anti-hepatitis drugs that perturb or inhibit viral

there is no other funding, and our institution does not fund others. This study was supported by the National Natural Science Foundation of China(No. 81460567) and the Inner Mongolia Natural Science Foundation of China (No. 2009MS1104, 2014MS0813 and 2019MS08198), and financial assistance was providedby the Department of Education of Inner Mongolia (No. NJ03148 and NJZY13244). The funds to support our research come from the project management organization. They are: National Natural Science Foundation of China, Inner Mongolia Science and Technology Department, Inner Mongolia Education Department. The funders had no role in study design, data collection and analysis, decision to publish, or preparation of the manuscript.

**Competing interests:** The authors have declared that no competing interests exist.

replication. However, their therapeutic effects are unsatisfactory. Consequently, the inadequately treated hepatitis infection is often prolonged and may culminate in cirrhosis or hepatocellular carcinoma [2]. These latter disorders are responsible for 1.4 million deaths per year [1]. Both control of infection, and liver cell injury, are strictly dependent upon protective immune responses, as hepatocyte damage is the price that the host must pay to rid itself of intracellular virions. This kind of immune liver injury is primarily caused by specific antiviral T cell-mediated cellular immunity [3]. Specifically, a large number of monocytes and lymphocytes infiltrate the parenchymal cells of the liver where Kupffer cells release many inflammatory cytokines, thereby inducing liver inflammation and leading to disrupted liver functions including dysfunction of drug metabolism [4, 5].

Relatively few studies have focused on the changes in metabolic enzymes and the associated regulatory mechanisms in viral hepatitis [6, 7]. Cytochrome P450 3A (CYP3A) is a predominant human liver monooxygenase responsible for metabolizing more than half of the drugs in use today [8]. Inflammatory stimuli and viral infections downregulate hepatic cytochrome P450 protein, which increases exposure to parent drugs by reducing their clearance and/or enhancing their bioavailability [5, 9]. As a result, the incidence of adverse reactions increases and drug treatment efficacy decreases. Effective hepatitis treatment requires the restoration of metabolic enzyme activity to normal levels and stabilization of liver metabolism. Although CYP3A has been shown to be down-regulated in hepatitis and liver cancer, the significance of this down-regulation, particularly as it pertains to changes in transcription, translation, and post-translational enzyme protein modifications in viral hepatitis, cirrhosis and hepatocellular carcinoma, remains uncharacterized [10, 11]. Additionally, NF-κB regulates inflammation, immune responses, and cell survival. It is also sensitive to oxidative stress and has been reported to participate in cell transformation, tumorigenesis, inflammation, and tissue injury [12]. Hence, activated NF-κB regulates various downstream inflammatory cytokines, oxidative enzymes, and iNOS, and affects liver injury [13–15]. Our previous work confirmed the activation of NF-κB during BCG-induced immune liver injury and its regulation of CYP2E1 [4, 5]. It is, therefore, highly plausible that NF-κB may regulate CYP3A in this process.

Sea buckthorn (*Hippophae rhamnoides* L.; Elaeagnaceae) is a spiny deciduous shrub that grows in > 10 provinces and autonomous regions in Northwest, Northeast, and Southwest China. As early as 1,400 years ago, sea buckthorn was widely used in Mongolian and Tibetan medicine [16–18]. It has been extensively consumed by various populations in Asia, the Nordic countries, and the Baltic region [19] and has been found to stimulate blood circulation, dissipate blood stasis, reduce phlegm, alleviate chest oppression, tone the spleen and stomach, promote body fluid production, relieve thirst, clear heat, and stop diarrhea [20]. Further, its fruit, bark, leaves, and other organs are rich in vitamins, carotenoids, flavonoids, essential oils, carbohydrates, organic acids, amino acids, and minerals [21]. Previous studies reported that *H. rhamnoides* has antioxidant, immunomodulatory, hepatoprotective, and anti-inflammatory effects [22, 23]. Sea buckthorn polysaccharide extract is antioxidant, anti-inflammatory, and antiviral, and also inhibits nuclear transcription factors, while promoting wound healing [24, 25]. A clinical study showed that sea buckthorn reduced the serum levels of laminin, hyaluronic acid, total bile acid, and collagen types III and IV in patients with liver cirrhosis, indicating that it may restrict the synthesis of collagen and other extracellular matrix components. Therefore, sea buckthorn may prove effective as a drug for prevention and treatment of liver fibrosis [26].

Since a primary therapeutic goal for hepatitis is to intervene in its inflammatory and oxidative nitrative stress pathways, we hypothesized that extracts of sea buckthorn may help normalize metabolic enzyme homeostasis, restore liver function, and target other pathological features of hepatitis. To test this hypothesis, we administered BCG to rats to induce an

immunological liver injury model and investigated the effects of HRP on liver function, inflammatory cytokines, such as TNF-α and IL-1β as well as on iNOS (nitrification kinase), CYP3A (drug-metabolizing enzyme), and NF-κB (nuclear transcription factor-κB). We examined the hepatoprotective effects of HRP and its molecular mechanisms in terms of liver function, inflammatory pathways, oxidative/nitrative stress, and CYP3A regulation, while also attempting to decipher the molecular mechanism of CYP3A transcription, translation and post-translational protein modification by *H. rhamnoides*. In this way, we assessed its value as a therapeutic agent in clinical applications.

## Materials and methods

### Experimental animals and reagents

Eight-week-old male Sprague-Dawley rats (body weight 270 ± 20 g; licensure: SCXK 2005–2001) were supplied by the Department of Laboratory Animal Science of Inner Mongolia University and maintained at 20 ± 1˚C under a 12-h light/dark cycle. The animals were housed in plastic cages with *ad libitum* food and water access. Animal experiments were approved by the Committee of Animal Ethics in Baotou Medical College, Baotou, Inner Mongolia, China, and conducted in accordance with the requirements of Chinese National Legislation. Sea buckthorn extract in fine granules were acquired from Sichuan Meidakang Pharmaceutical Co., Ltd. Sichuan, China. *Mycobacterium bovis* BCG vaccine (60 mg; Batch No. 2015–1) and MDZ reference substance (purity: 99.5%) were acquired from the National Institute for the Control of Pharmaceutical and Biological Products, Beijing, China. Injectable MDZ (No. H10980025) was provided by Jiangsu Enhua Pharmaceutical Co. Ltd., Jiangsu, China. Kits for cytoplasmic and nuclear protein extraction, bicinchoninic acid (BCA) protein assay (No. AR0146), sodium dodecyl sulfate polyacrylamide gel electrophoresis (SDS-PAGE), rat IL-1β ELISA (No. BA2913), TNF-α ELISA (No. BA0527), and rabbit polyclonal antibodies against CYP3A (No. A00339), iNOS (No. EK0394), LMNA(Lamin A/C) (Cat. No. BA1227), and GAPDH (No. BA2913) were obtained from Wuhan Boster Biological Engineering Co. Ltd., Wuhan, China. Alanine transaminase (ALT) and serum aspartate transaminase (AST) levels were measured with commercial kits obtained from Beijing Bei Hua Kang Tai Clinical Reagent Co., Beijing, China. The NADPH regeneration system was purchased from Beijing HuiZhiTaiKang Pharmaceutical Technology Co. Ltd., Beijing, China (Batch Nos. NRS(A)-180906 and NRS(B)-180906). The total RNA Extraction kit, cDNA Synthesis kit and 2X Realtime PCR Master Mix were purchased from Nanjing Bordi Biotechnology Co., Ltd. Nanjing, China.

### Animal grouping and preparation of immunological liver injury models

The rats were weighed, numbered, and randomly divided into six groups of 16 rats each as follows: control (intravenous normal saline (10 mL/kg) injection); HRP group (HRP) (oral HRP administered at medium dose (100 mg/kg, twice daily for 13 d); immune-mediated liver injury group (BCG) (single intravenous 125 mg/kg BCG injection); BCG + HRP group (BCG + HRP) (intravenous injection of 125 mg/kg BCG on the first day followed by oral HRP administered at low (50 mg/kg), medium (100 mg/kg), and high (200 mg/kg) doses (BCG +HRPL, BCG+HRPM, and BCG+HRPH, respectively) twice daily for 13 d. Ten rats from each group were sacrificed by decapitation at the end of the experiment. Livers and spleens were excised and weighed. Liver tissues were immersed in formaldehyde-alcohol stationary liquid (formaldehyde:alcohol = 1:9) prepared in advance. Some of the liver tissue sections were embedded in paraffin for hematoxylin and eosin (H&E) staining and observation under an Olympus CX23 microscope (Olympus, Tokyo, Japan); while others were frozen in liquid nitrogen for western blotting, metabolic activity, RT-PCR, and ELISA assays. Sera were sampled to

determine aminotransferase levels. The remaining six rats in each of the six groups were used to measure the metabolic activity of CYP3A by HPLC at the end of the experiment [27].

## Serum ALT and AST determination

Sera were collected 14 d after BCG injection and serum ALT and AST were measured with an automated biochemical analyzer (Hitachi 7600–020, Tokyo, Japan).

## In vivo CYP3A metabolic activity measurement by HPLC

To further investigate the metabolic activity of CYP3A in the body, at the end of the experimental period, the remaining six rats in the control-, HRP-, BCG-, and BCG + HRPM groups were intravenously injected once with MDZ (10 mg/kg). Blood from the angular vein (0.2 mL) was collected after 2, 5, 10, 15, 20, 30, 60, 120, and 240 min [28, 29], placed in heparinized tubes, and centrifuged at 4°C for 5 min at 3,000 × $g$. Plasma was collected and stored at -20°C until further analysis. According to a previously described method [11], plasma concentration-time curves were plotted and the pharmacokinetic parameters ($A$, $AUC$, $T_{1/2}$, $Ke$, and $CL$) for MDZ and 1-hydroxymidazolam (1-OH-MDZ) were measured and fitted using DAS v.3.0 (Shanghai Bojia Pharmaceutical Technology Co., Ltd., Shanghai, China). CYP3A metabolic activity was evaluated from the concentration-time curves and pharmacokinetic parameters for MDZ and 1-OH-MDZ.

## In vitro CYP3A metabolic activity measurement by HPLC

Next, to evaluate the metabolic activity of liver CYP3A, liver microsomes were prepared by calcium precipitation [30]. The liver tissue was rinsed repeatedly with frozen Tris buffer until a khaki color was observed. Liver tissue (1 g) was placed in cold phosphate buffer (pH 7.4). Liver homogenate was prepared using an internal cutting homogenate mechanism (0.2 min; 25°C; 3,500 × $g$). The liver homogenate was centrifuged at 4°C for 20 min at 12,000 × $g$. Each milliliter of supernatant fraction was combined with 0.1 mL CaCl$_2$ (88 mM) and gently shaken at 0–4°C for 5 min. The mixture was transferred to a centrifuge tube and spun at 15,000 × $g$ and 4°C for 30 min. The supernatant was discarded and the sediment was resuspended in phosphate buffer and centrifuged at 15,000 × $g$ and 4°C for 30 min. The resultant pink precipitate served as the source of liver microsomes, which were resuscitated in phosphate buffer and divided into two aliquots. The first was stored at -80°C until further analysis and the second was used to determine the protein content. Microsomal protein content were measured by BCA assay. The reaction volume was 200 μL, to which 100 μL of hepatic microsomes, probe drugs, and Tris-HCl buffer were added. The mixture was incubated at 37°C for 5 min and NADPH was added to initiate the reaction. The mixture was incubated for 45 min and a triple volume of ethyl acetate at 0°C was added to terminate the reaction. The mixture was vortexed for 5 min and centrifuged at 4°C for 10 min at 8,000 × $g$. A 2.5-mL supernatant aliquot was removed and dried by nitrogen gas stream at 40°C. The residue was dissolved in 300 μL of the mobile phase and the solution was injected into the HPLC system for analysis [31, 32]. The metabolic activity of CYP3A was assessed using the rates of 1-OH-MDZ production (metabolism) from MDZ [11].

## Western blot analyses of CYP3A, iNOS, and NF-κB

Liver tissues were weighed and their proteins were prepared with a protein extraction kit according to the manufacturer's instructions. The proteins were quantitated by BCA assay. Equal amounts of protein (30 μg) were separated on 5–10% (w/v) polyacrylamide gel and electrophoretically transferred onto polyvinylidene difluoride (PVDF) membranes. Nonspecific

sites were blocked with Tris-buffered saline containing 5% (w/v) skim milk for 1 h at 25˚C. The membranes were incubated with rabbit polyclonal antibodies against CYP3A, iNOS, and NF-κB at 1:200 dilution in blocking solution for 2 h at 25˚C, followed by incubation with goat anti-rat IgG at 1:2,000 dilution in blocking solution for 30 min at 25˚C. The membranes were then washed with PBS and proteins were visualized by enhanced chemiluminescence according to the manufacturer's instructions (AR1190; Wuhan Boster Biological Engineering Limited Company, Wuhan, China) and photographed with an OmegaLum C imaging system (Gel Company, San Francisco, CA, USA). An anti-GAPDH polyclonal antibody was the internal standard for CYP3A and iNOS. LMNA rabbit polyclonal antibody was the internal standard for NF-κB.

## RNA extraction, cDNA synthesis and quantitative RT-PCR

To explore the effect of HRP on CYP3A transcription, the mRNA of CYP3A was detected by RT-PCR. Total RNA was prepared using RNA fast200 Extraction kit according to the manufacturer's instructions. RNA concentration was determined by measuring absorbance at 260 nm. RNA purity and integrity were confirmed via formaldehyde-agarose gel electrophoresis. Visualization was conducted with ethidiumbromide. cDNA synthesis from total RNA was performed using the First-Strand cDNA Synthesis kit. Real-time RT-PCR was used to measure the relative expression of CYP3A mRNA, using the ABI 7900 HT Real-Time PCR System and SyBr® Green Master Mix reagent as described. The forward primer for the amplification of CYP3A was 5ʹ–TCCTGGCTACCATCCTCGTG–3ʹ, and the reverse primer was 5ʹ–CCACTGG TGAAGGTGGGAGA–3ʹ. The GAPDH primers used were 5ʹ– AATGGTGAAGGTCGGTGTG AACG–3ʹ (forward) and 5ʹ– TCGCTCCTGGAAGATGGTGATGG–3ʹ (reverse). Values were normalized to GAPDH mRNA using the $2^{-\Delta\Delta Ct}$ method and the expression level in control samples was arbitrarily set to 1.

## IL-1β and TNF-α levels in rat liver measured by ELISA

To explore the effect of HRP on inflammatory cytokines, ELISA was used to detect IL-1β and TNF-α. Liver tissue (0.2 g) was sliced, added to PBS (1 mL; pH 7.4), homogenized and centrifuged at 4˚C for 20 s at $3,500 \times g$, after which the supernatant was collected. Absorbances were measured within 30 min in a microplate reader (680; Bio-Rad Laboratories, Hercules, CA, USA) at 450 nm according to the kit manufacturer's instructions. The 3,3ʹ,5,5ʹ-tetramethyl-benzidine (TMB) reagent was used as the blank control. The optical densities (OD) of the TMB reagent were subtracted from those of the standard and samples. The standard curve and regression equation were then plotted and the concentrations of IL-1β and TNF-α in the liver homogenates were interpolated from the standard curve equation.

## Statistical analysis

Statistical analyses were performed using SPSS v. 19.0 (IBM Corp., Armonk, NY, USA). Data are presented as means ± standard deviation (SD). Treatment means were compared by one-way analysis of variance (ANOVA) followed by Tukey's test. $P < 0.05$ was considered statistically significant.

## Results

### Changes in the pathological manifestations and biochemical indices of rat liver

Five to seven days following injection of BCG into tail veins, rats became aggressive and irritable, causing them to bite each other. Two weeks later, H&E-stained liver sections were

observed microscopically (Fig 1A–1F). Hepatic lobules in the control group presented with a normal structure and the liver cells were round and normal in size with clear nucleoli. Only those in the portal area presented with slight inflammatory infiltration (Fig 1A). The livers from the HRP group showed normal morphological characteristics, similar to the control group (Fig 1B). Alternatively, in the BCG group, the liver parenchyma and portal area were surrounded by numerous monocytes and lymphocytes, indicative of inflammation, and the cell masses varied in size and were diffuse. Consequently, the hepatic cord structure was unclear. The liver cells presented with vacuolar degeneration (Fig 1C). Relative to the

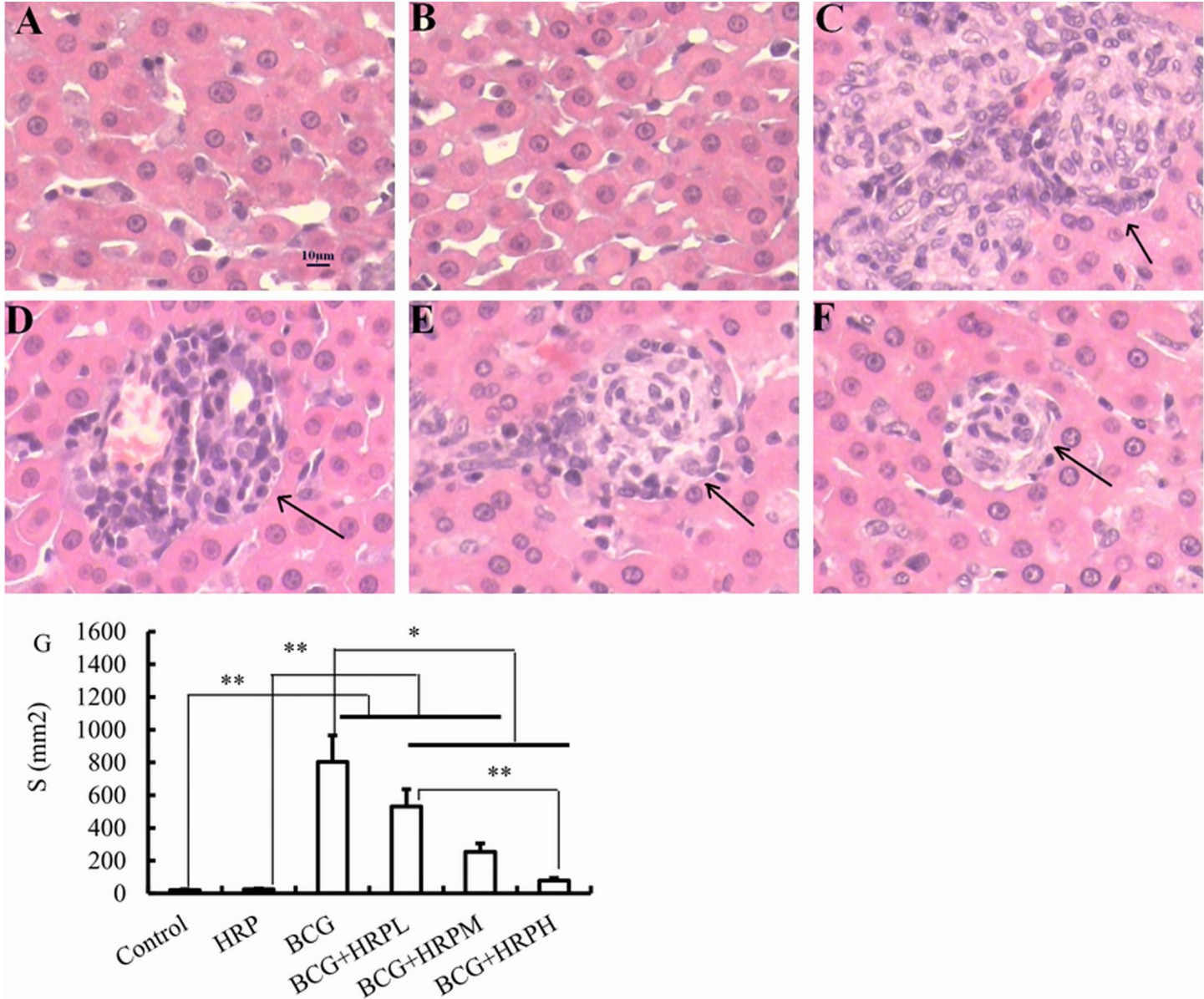

**Fig 1. Effects of *Hippophae rhamnoides* L. (HRP) on histopathological or biochemical changes in rat liver.** (A–F): Representative light microscope images of rat liver (hematoxylin staining; original magnification, ×400) for control (A); HRP (B); BCG group (C; arrows in figure point to granulomatous mass composed of inflammatory cells); BCG + HRPL (50 mg kg$^{-1}$, D); BCG + HRPM (100 mg kg$^{-1}$, E);and BCG + HRPH (200 mg kg$^{-1}$, F). Representative light microscope images of inflammatory cell infiltrate area (S) in liver for each group is shown in G). Data are presented as means ± SD of $n$ = 10 rats. Asterisks stars above bars indicate differences between groups.

untreated, liver histology was significantly improved by the addition of HRP following BCG stimulation, in a dose-dependent manner (Fig 1D–1F). The representative light microscope images of liver inflammatory cell infiltrate areas (S) for each group are shown in Fig 1G.

Liver and spleen weight significantly ($P < 0.05$) increased in the BCG group compared with the control. However, HRP treatment suppressed liver and spleen weight gain in a dose-dependent manner following BCG stimulation ($P < 0.05$; Table 1). AST and ALT levels were significantly ($P < 0.05$) elevated in the BCG group relative to the control. However, the HRP treatment in BCG-stimulated rats caused a significant ($P < 0.05$) decrease in AST and ALT compared with the control (Fig 2A and 2B). Similarly, HRP intervention in normal rats resulted in liver weight, spleen weight, and transaminase levels to be similar to those in the control group.

## Changes in CYP3A metabolic activity

The specifics of the *in vitro* and *in vivo* methods employed for detecting CYP3A metabolic activity by HPLC are shown in Fig 3. The retention times were 7.216 min for 1-hydroxymidazolam and 9.193 min for midazolam. The chromatographic peaks were symmetric and there was no interference from endogenous material. The linear correlation was strong in the concentration range of 0.1–20 mg/L (y = 116946x+81281, $r = 0.997$) and fully met the experimental requirements.

MDZ, a drug metabolized by CYP3A, was intravenously administered and the plasma concentration-time curves for the control-, HRP-, BCG-, and BCG + HRPM groups (100 mg/g) are shown in Fig 4. The plasma MDZ concentrations were measured by HPLC. Plasma MDZ was higher in the BCG group compared to the control or HRP groups at 5, 10, 15, 20, 30, 60, and 120 min post-injection. For the BCG + HRPM group, however, the plasma MDZ was significantly ($P < 0.05$) lower than that of the BCG group at 5, 10, 15, and 20 min, and was significantly higher than that for the control and HRP groups at 5, 10, and 15min (Fig 4). Compared with the control group, the plasma concentration of MDZ in HRP group rats at each time did not change significantly.

**Table 1. Effects of HRP on liver, spleen weight relative to body weight ratio (%), in *Bacillus Calmette-Guérin* (BCG)-induced hepatic injury in rats *in vivo*.**

| Group | Liver weight/body weight (%) | spleen weight /body weight (%) |
|---|:---:|:---:|
| Control | 5.0 ± 0.6 | 0.4 ± 0.1 |
| HRP | 5.1 ± 0.6 | 0.3 ± 0.1 |
| BCG | 8.6 ± 0.5\*\*,▲▲ | 1.3 ± 0.2\*\*,▲▲ |
| BCG+HRPL | 8.0 ± 1.1\*,▲ | 1.2 ± 0.2\*\*,▲▲ |
| BCG+HRPM | 7.5 ± 0.5\*,▲ | 1.0 ± 0.1\*,▲,#,△ |
| BCG+HRPH | 7.1 ± 0.6\*,▲,#,△ | 0.8 ± 0.1\*,▲,#,△△ |

Data are presented as means ± SD of $n = 10$ rats.

\*$P < 0.05$,

\*\*$P < 0.01$ *vs.* control;

▲$P < 0.05$,

▲▲$P < 0.01$ *vs.* HRP;

#$P < 0.05$,

##$P < 0.01$ *vs.* BCG group;

△$P < 0.05$,

△△$P < 0.01$ *vs.* BCG + HRPL group.

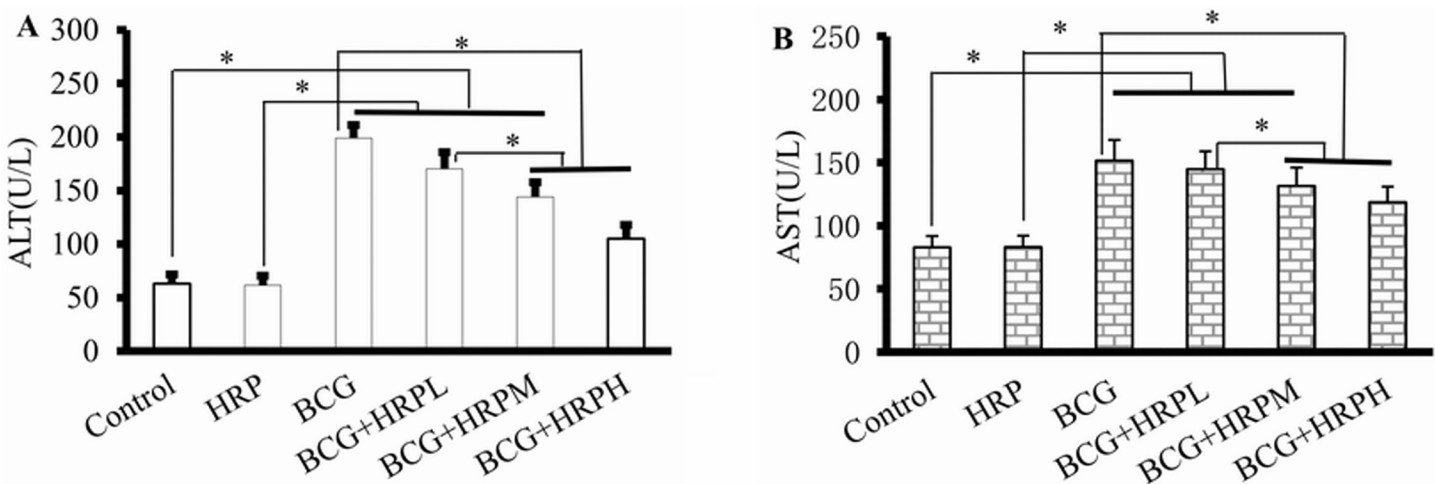

**Fig 2.** Effects of HRP on serum alanine aminotransferase (ALT, A) and aspartate aminotransferase (AST, B) in *Bacillus Calmette-Guérin* (BCG)-induced hepatic injury in rats *in vivo*. Data are presented as means ± SD of *n* = 10 rats. Asterisks stars above bars indicate differences between groups.

The MDZ pharmacokinetic parameters of each group are shown in Table 2. Results show that the *A*, $T_{1/2}$, and $AUC_{(0-t)}$ of the BCG group were significantly higher than those of the control and HRP groups, while *CL* was significantly lower than the control or HRP group ($P < 0.05$). Similarly, the *A*, $T_{1/2}$, and $AUC_{(0-t)}$ of the BCG+HRPM group were significantly higher than those of the control group or the HRP group and lower than the BCG group; while

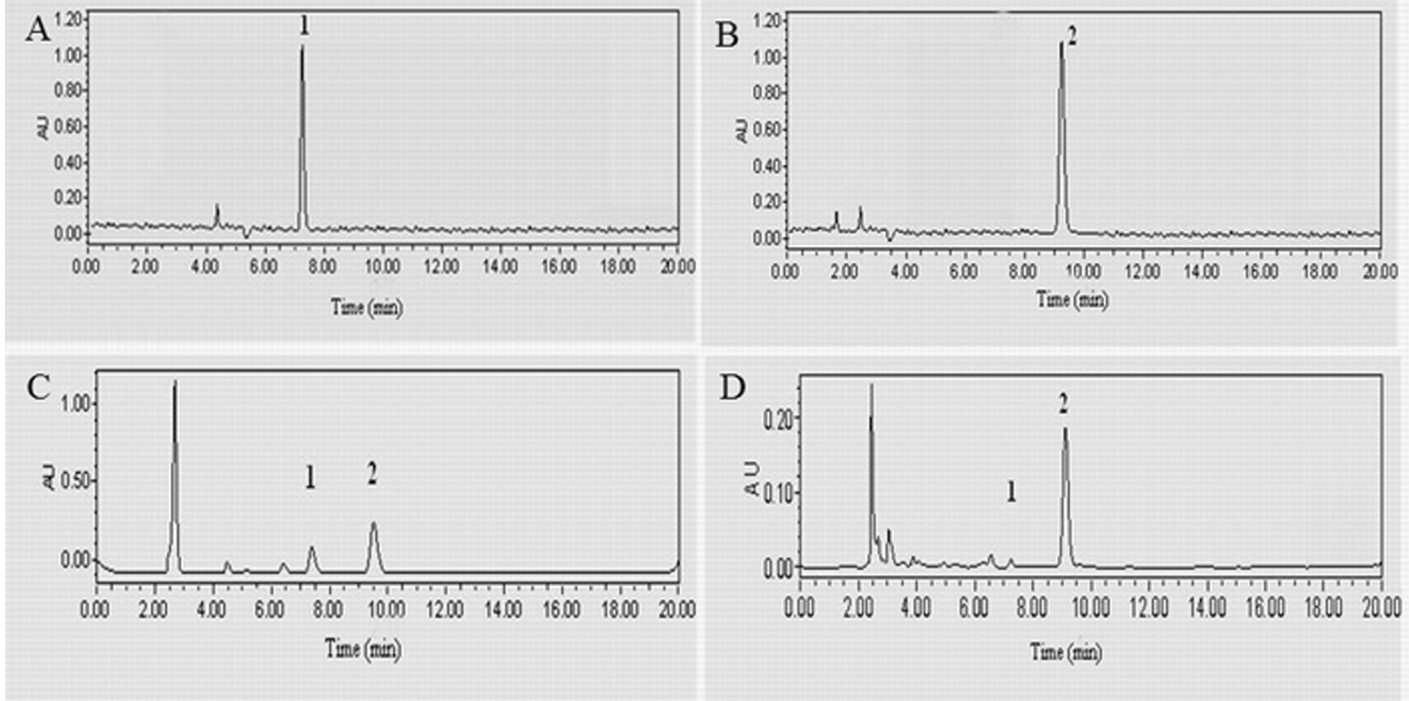

**Fig 3. HPLC chromatograms.** (A) 1-hydroxymidazolam, (B) midazolam, (C) microsomes spiked with midazolam for incubation, (D) plasma sample obtained from rat after intravenous midazolam injection; 1, 1-hydroxymidazolam; 2, midazolam.

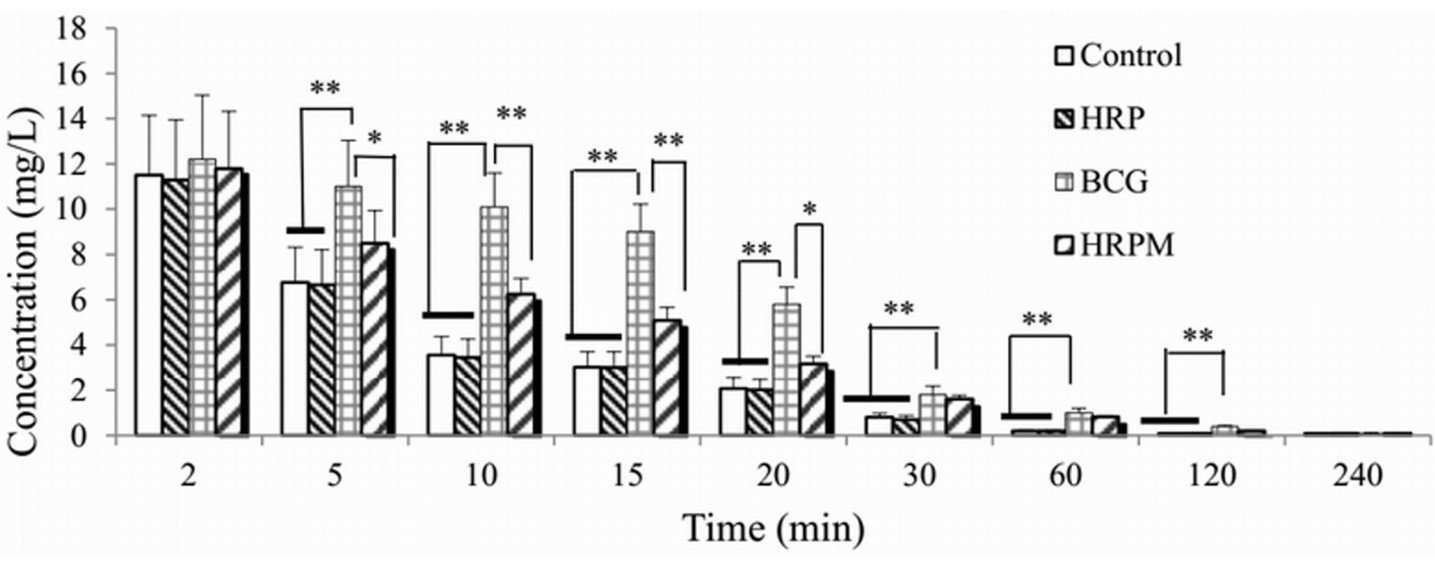

**Fig 4. Plasma concentration-time profiles of midazolam (10 mg kg$^{-1}$) following its intravenous injection in control, BCG-stimulated-, HRP-, and BCG + HRPM-treated rats.** Blood samples were collected at 2, 5, 10, 15, 20, 30, 60, 120, and 240 min after midazolam administration. Plasma midazolam was measured as described in the Materials and Methods section. Data are means ± SD, $n$ = 6, asterisks stars above bars indicate differences between groups.

*CL* was significantly lower than the control group or HRP group and higher than the BCG group. ($P < 0.05$). *Ke* did not change significantly in each group.

The drug time curve and pharmacokinetic parameters of 1-hydroxymidazolam, a metabolite of MDZ catalyzed by CYP3A, are shown in Fig 5. Plasma 1-hydroxymidazolam in the BCG group was lower than that in the control or HRP at 2, 5, 10, 15, 20, and 30 min. For the BCG + HRPM group, however, the plasma 1-hydroxymidazolam was significantly higher than that in the BCG group and lower than that in the control or HRP at 2, 5, 10, 15, 20, and 30 min ($P < 0.05$) (Fig 5). Similar to the drug time curve of the prototype drug, the blood concentration

**Table 2. Pharmacokinetic parameters of midazolam (10 mg kg$^{-1}$) following its intravenous injection in control, BCG-stimulated-, HRP-, and BCG + HRPM-treated rats.**

|  | Control | HRP | BCG | BCG+HRPM |
|---|---|---|---|---|
| *A* mg/L | 9.11 ±1.91 | 9.03 ±1.83 | 16.46 ±3.29*\*,▲▲ | 11.13 ±2.45*,▲,## |
| $T_{1/2}$ min | 8.93 ±1.88 | 8.87 ±1.84 | 11.97 ±2.39*,▲▲ | 9.94 ±2.63*,▲ |
| *Ke* 1/min | 0.08 ±0.02 | 0.08 ±0.02 | 0.06 ±0.01 | 0.07 ±0.01 |
| *CL* L/min/kg | 1.10 ±0.23 | 1.06 ±0.21 | 0.61 ±0.12*\*,▲▲ | 0.90 ±0.20*,▲,# |

Parameters were calculated from the plasma midazolam concentration profiles (see Fig 5) as described in the Materials and Methods section. Data are presented as means ± SD of $n$ = 6 rats.

\* $P < 0.05$,

\*\* $P < 0.01$ *vs.* control;

▲ $P < 0.05$,

▲▲ $P < 0.01$ *vs.* HRP;

# $P < 0.05$,

## $P < 0.01$ *vs.* BCG group;

△ $P < 0.05$,

△△ $P < 0.01$ *vs.* BCG + HRPL group. *A*, total drug in the body; *AUC*, area under the curve; *Ke*, elimination rate constant; $T_{1/2}$, half-life; *CL*, clearance rate.

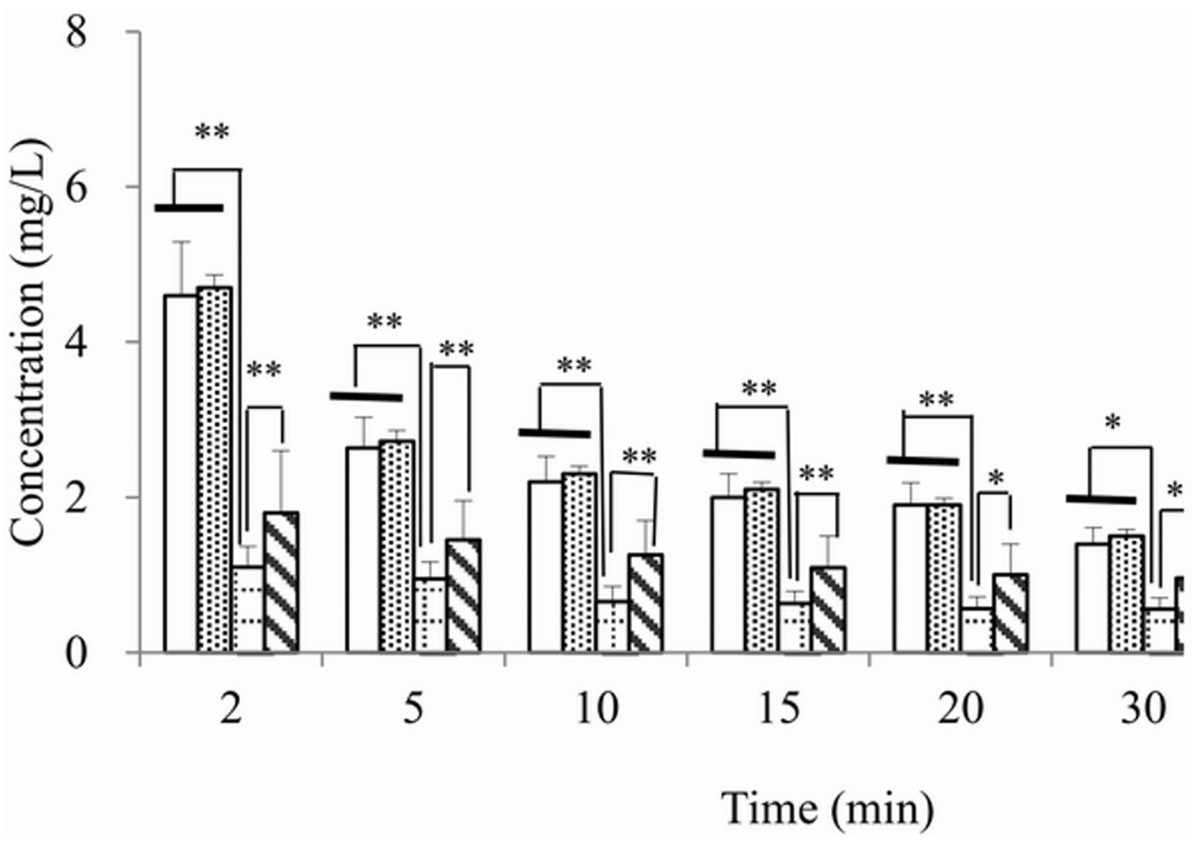

**Fig 5. Plasma concentration-time profiles of 1-hydroxymidazolam after intravenous injection in control, HRP-, BCG-stimulated-, and BCG + HRPM-treated rats midazolam (10 mg kg$^{-1}$).** Blood samples were collected at 2, 5, 10, 15, 20, 30, 60, 120, and 240 min after midazolam administration. Plasma midazolam was measured as described in the Materials and Methods section. Data are means ± SD, $n = 6$, asterisks stars above bars indicate differences between groups.

of 1-OH-MDZ in the HRP group and control group rats did not change significantly at the various time points. The 1-OH-MDZ pharmacokinetic parameters of each group are shown in Table 3. The $A$, $C_{max}$, $CL$ and $AUC_{(0-t)}$ of the BCG group were significantly lower than those of the control group or the HRP group, while $T_{1/2}$ was significantly higher than the control or HRP group ($P < 0.05$). The $A$, $C_{max}$, $CL$ and $AUC_{(0-t)}$ of the BCG+HRPM group were significantly lower than those of the control and HRP groups, and higher than the BCG group, while $T_{1/2}$ was significantly higher than the control and HRP group and lower than the BCG group ($P < 0.05$). $Ke$ did not differ significantly between groups. The above data suggests that HRP does not induce or inhibit CYP3A metabolic activity in normal rats. However, it did inhibit the down-regulation of CYP3A metabolic activity *in vivo* in rats induced by BCG.

We used liver microsomes to catalyze the CYP3A probe drug MDZ in 1-OH-MDZ *in vitro* experiments. The 1-OH-MDZ / MDZ ratio determined after reaction reflects the metabolic activity of liver CYP3A. *In vitro* assays confirmed that HRP attenuated CYP3A down-regulation resulting from BCG-induced immunological liver injury in a dose-dependent manner (Fig 6). Further, immuno-stimulation induced by BCG down-regulated 1- OH-MDZ metabolism. In contrast, HRP reduced this repression in a dose-dependent manner following BCG stimulation ($P < 0.05$). In the HRP and control groups, the 1-OH-MDZ metabolic rate did not change significantly, suggesting that HRP does not induce or inhibit liver CYP3A metabolic

**Table 3. Pharmacokinetic parameters of 1-hydroxymidazolam after intravenous injection in control, HRP-, BCG-stimulated-, and BCG + HRPM-treated rats midazolam (10 mg kg$^{-1}$).**

| Parameter | Control | HRP | BCG | BCG+HRPM |
|---|---|---|---|---|
| $A$ mg/L | 9.35 ±1.72 | 9.44 ±1.68 | 0.73 ±0.17**,▲▲ | 0.895 ±0.08*,▲ |
| $T_{1/2}$ min | 55.72 ±6.93 | 56.81 ±6.87 | 80.16 ±10.26* | 66.51 ±7.74*,▲,# |
| $Ke$ 1/min | 0.04 ±0.01 | 0.03 ±0.01 | 0.03 ±0.01 | 0.03 ±0.01 |
| $C_{max}$ mg/L | 2.20 ±0.25 | 2.31 ±0.28 | 0.66 ±0.15**,▲▲ | 1.26 ±0.25*,▲,# |
| $CL$ L/min/kg | 1.10 ±0.16 | 1.14 ±0.17 | 0.61 ±0.11**,▲▲ | 0.90 ±0.12*,▲,# |
| $AUC_{(0-t)}$ mg/L·min | 146.20 ±13.31 | 148.35 ±13.69 | 102.92 ±13.93* | 121.75 ±13.44*,▲,# |

Parameters were calculated from the plasma 1-hydroxymidazolam concentration profiles (see Fig 5) as described in the Materials and Methods section. Data are presented as means ± SD of $n$ = 6 rats.

* $P < 0.05$,

** $P < 0.01$ *vs.* control;

▲ $P < 0.05$,

▲▲ $P < 0.01$ *vs.* HRP;

# $P < 0.05$,

## $P < 0.01$ *vs.* BCG group;

△ $P < 0.05$,

△△ $P < 0.01$ *vs.* BCG + HRPL group. $A$, total drug in the body; $T_{1/2}$, half-life; $Ke$, elimination rate constant; $C_{max}$, peak concentration; $CL$, clearance rate; $AUC$, area under the curve.

activity in normal rats. However, it was found to prevent the down-regulation of liver CYP3A metabolic activity *in vitro* in rats induced by BCG.

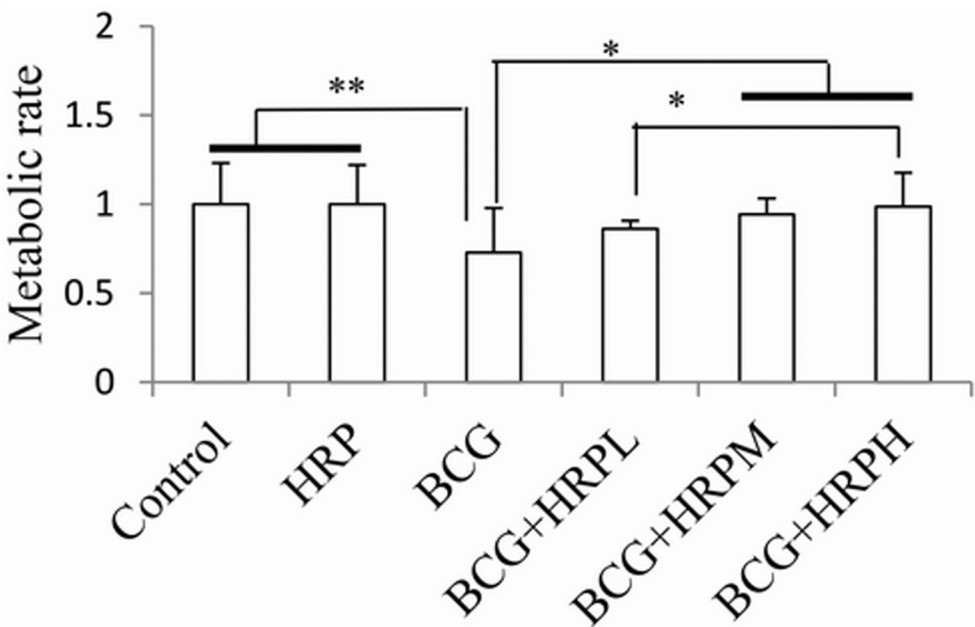

**Fig 6. Effect of HRP on CYP3A metabolic activity in microsomes from rat liver following immune-mediated injury.** Metabolic rate was indicated by the ratio of 1-hydroxymidazolam to midazolam after *in vitro* incubation with microsomes and weighted by the control group. Data represent means ± SD of $n$ = 10 rats. Asterisks stars above bars indicate differences between groups.

## Changes in NF-κB, iNOS, and CYP3A expression

Western blot analysis showed uniform density among bands compared with the internal reference LMNA or GAPDH proteins. Thus, protein quantitation was accurate and the amounts of proteins in the samples were consistent (Fig 7). NF-κB and iNOS were significantly upregulated in the BCG group compared with the control or HRP. However, HRP pretreatment suppressed NF-κB and iNOS overexpression in a dose-dependent manner (Fig 7A and 7B; $P <$ 0.05). As shown in Fig 7C, CYP3A was expressed at basal levels in normal liver tissue. BCG injection significantly down-regulated CYP3A, whereas HRP pretreatment alleviated this effect in a dose-dependent manner ($P < 0.05$). Compared with the control group, the expression of NF-κB, iNOS, and CYP3A in HRP group did not change significantly. These results suggest that HRP inhibits CYP3A down-regulation in the liver of rats with BCG-induced immune liver injury, while eliciting no significant effect on CYP3A expression in normal rats.

## The effect of HRP on *CYP3A* mRNA expression in rat liver

Our results show that BCG significantly reduces CYP3A mRNA expression in rat liver ($P <$ 0.05), while HRP prevents this down-regulation during immune liver injury in a dose-response manner. Compared with the control group, no significant change was observed in CYP3A mRNA in HRP rats, indicating that HRP has no effect on CYP3A transcription in normal rats. These findings suggest that HRP inhibits CYP3A down-regulation at the level of transcription during immune liver injury. Combined with the western blot results, we further supposed that HRP may be involved in BCG-mediated down-regulation of CYP3A transcription levels by inhibiting NF-κB (Fig 8).

## Effect of HRP on TNF-α and IL-1β content in immune-mediated rat liver injury

As shown in Fig 9, rats injected with BCG for 14 d presented with significantly upregulated proinflammatory cytokines (TNF-α and IL-1β) relative to the control. However, HRP

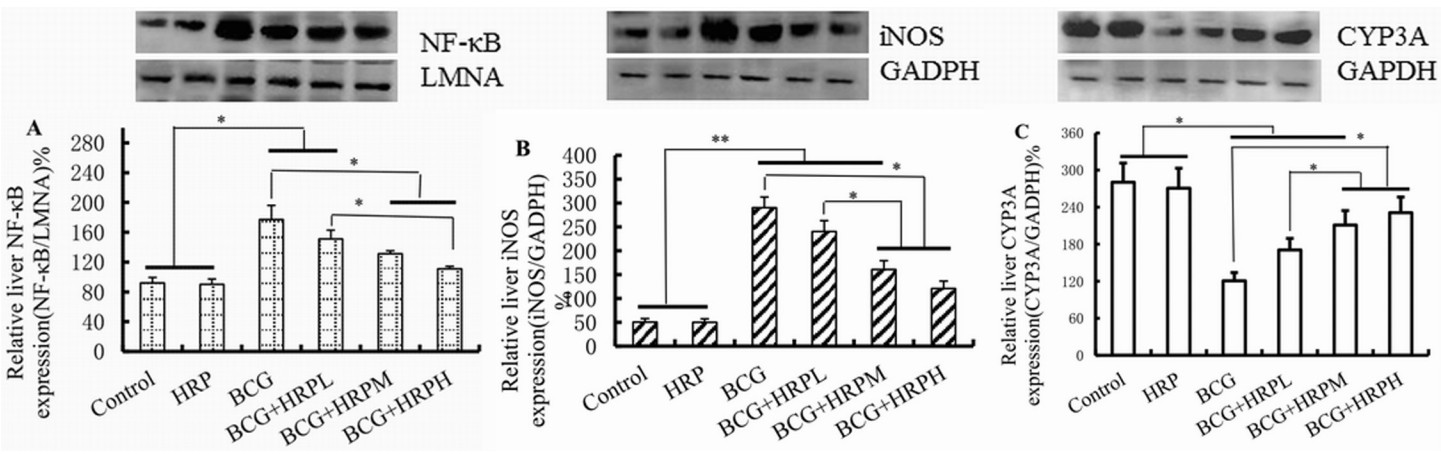

**Fig 7. Effects of HRP on NF-κB, iNOS, and CYP3A expression in rat livers following immune-mediated injury.** Rats were treated with BCG (125 mg kg$^{-1}$, intravenously, once for 2 wks) or BCG + HRP (50, 100, or 200 mg kg$^{-1}$ d$^{-1}$, oral administration for 13d). Liver proteins were extracted to determine NF-κB, iNOS, and CYP3A expression. Equal amounts (30 μg) of protein were subjected to SDS-PAGE followed by western blot analysis using anti-NF-κB, anti-iNOS, and anti-CYP3A antibodies. Results were normalized to LMNA or GAPDH. NF-κB (A), iNOS (B), and CYP3A (C) protein expression levels in rat liver were measured by western blot. NF-κB, iNOS, and CYP3A expression levels were quantified by ImageQuant software (GE Healthcare Life Science, Little Chalfont, UK). Data represent means ± SD of three independent experiments. Asterisks stars above bars indicate differences between groups.

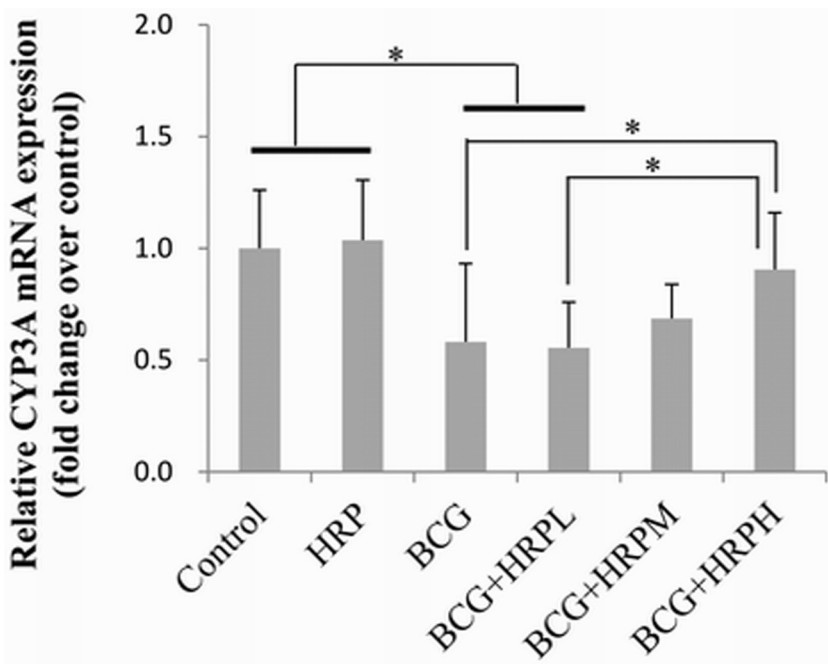

**Fig 8. Effect of HRP on CYP3A mRNA expression according to GAPDH levels in rat livers of immune-mediated liver injury.** Rats were treated with BCG (125 mg kg-1, intravenously, once for 2 wks) or BCG + HRP (50, 100, or 200 mg•kg$^{-1}$•d$^{-1}$, oral administration for 13d). The mRNA expression of CYP3A was measured by real-time PCR. Each bar represents the mean ± SD from three independent experiments (n = 10/group). Asterisks stars above bars indicate differences between groups. The significance of the data was determined by one-way analysis of variance (ANOVA) followed by Tukey's test.

pretreatment repressed TNF-α and IL-1β upregulation in a dose-dependent manner ($P <$ 0.05). Compared with the control group, no significant change was observed in the TNF-α and IL-1β levels in the HRP group.

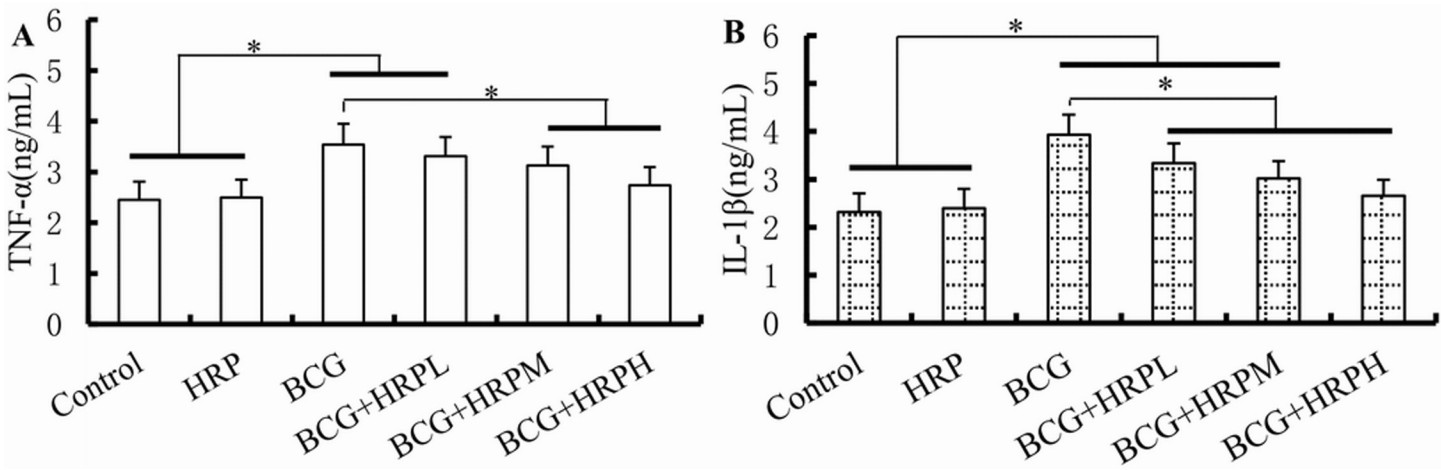

**Fig 9. Effects of HRP on TNF-α and IL-1β content in rat liver after immune-mediated injury.** Rats were treated with BCG (125 mg kg$^{-1}$, intravenously, once for 2 wks) or BCG + HRP (50, 100, or 200 mg kg$^{-1}$ d$^{-1}$, oral administration for 13d). Each bar represents the mean ± SD of three independent experiments. $n$ = 10/group, asterisks stars above bars indicate differences between groups.

### The schematic abstract of the mechanism

As shown in Fig 10, rats injected with BCG for 14 d presented with significantly upregulated TNF-α, IL-1β, iNOS and NF-κB relative to the control. However, HRP pretreatment repressed TNF-α, IL-1β, iNOS and NF-κB upregulation in a dose-dependent manner. This effect might occur at the transcriptional level and the level of nitration modification after enzyme protein translation.

## Discussion

Our study found that HRP alone had no effect on liver drug enzyme content, liver CYP3A protein expression, transcription or metabolic activity, indicating that HRP does not induce or inhibit CYP3A in this state. Alternatively, BCG immune-stimulation downregulates liver CYP3A; while HRP intervention reverses this effect in a dose-dependent manner and protect the liver's metabolic function. CYP3A is a monooxygenase primarily expressed in the liver, and oxidizes and metabolizes more than half of the drugs used clinically [8]. Changes in its metabolic activity determine the blood concentration of the metabolized drug in the body, causing drug interactions or drug toxicity reactions, thereby directly affecting the treatment effect [9]. MDZ was recommended by the Food and Drug Administration (FDA) as an *in vitro* and *in vivo* probe for CYP3A activity in rats and humans [29]. Therefore, MDZ effectively evaluates hepatic CYP3A metabolic activity in response to immunological liver injury, directly influencing therapeutic efficacy and drug-drug interactions. Herein, we describe an alternate mechanism for HRP's hepatoprotective effect, which is to regulate the unbalanced CYP3A during immune liver injury, restore the liver's metabolic capacity, and eliminate harmful substances through metabolism. According to this study, the mechanism of HRP regulation of CYP3A is as follows:

First, HRP reverses CYP3A down-regulation during immune liver injury at the transcriptional level. Direct experimental evidence indicates that the mRNA expression of CYP3A becomes down-regulated, while NF-κB is upregulated during BCG immune liver injury. However, HRP administration inhibits NF-κB and reverses CYP3A down-regulation, suggesting

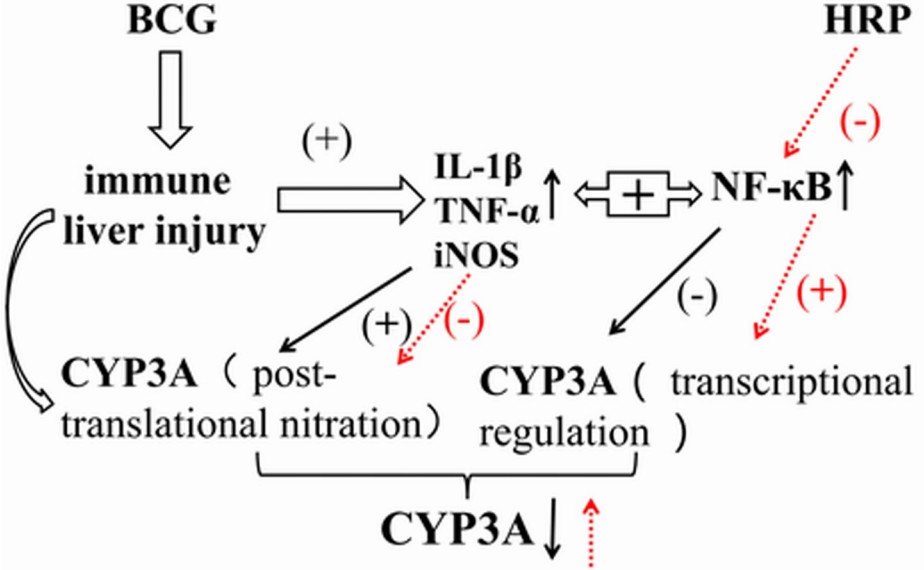

**Fig 10. The schematic abstract of the mechanism.**

that NF-κB plays a key role in CYP3A regulation. Our study as well as those of others [9] found that during BCG-induced immune liver injury, a large number of inflammatory cells infiltrated the liver, while Kupffer cells released high levels of IL-1β and TNF-α, and activated NF-κB. Immunological liver injury is an immune response [4, 5, 11]. Liver cells become damaged during immune-induced clearance of viruses, which serves as an integral component in the pathogenesis of viral hepatitis, cirrhosis and hepatocellular carcinoma [33, 34]. While a normal immune response has a protective effect on the body, an excessively strong immune response results in immune damage, multiple organ failure, and eventually death, as was evidenced by the SARS virus in 2003 as well as the coronavirus outbreak (2019-nCoV) in 2019, the severity of which is closely related to the inflammatory storm induced by virus infection [35–37].

We also found that during BCG-induced immune liver injury, a large amount of iNOS is induced which catalyzes the production of large amounts of NO, generates nitrifying stress [4], and activates NF-κB since NF-κB is more sensitive to inflammatory reactions and oxidative nitrification [5]. Importantly, an NF-κB activation site is located between the CYP3A promoter nucleotide sequences 2326 and 2297, and hence, NF-κB inhibits the transcription of CYP3A [38, 39]. It also binds to fixed nucleotide sequences in the gene promoter regions of several cytokines, inflammatory mediators, and transcription factors and, thus, plays important roles in immune and inflammatory responses [31, 40, 41]. Conversely, HRP inhibits the release of inflammatory cytokines (IL-1β and TNF-α) and inhibits iNOS induction, thereby inhibiting NF-κB activation and promoting CYP3A transcription. Hence, although BCG immune stimulation induced CYP3A protein down-regulation, HRP administration dose-dependently reversed this down-regulation and protected liver metabolic function, which may have been caused by HRP promoting CYP3A transcription.

Secondly, we found that the protective effect of HRP on CYP3A in immunological liver injury rats occurred at the post-translational modification level. We observed disproportionate down-regulation of CYP3A protein, mRNA and metabolic activity levels, suggesting the existence of post-translational modification of the enzyme protein. During immunological liver injury, a large amount of iNOS is induced, which catalyzes NO production by guanidino nitrogen on L-arginine [4], accounting for the molecular source of nitrification during CYP3A post-translational modification [42]. Both iNOS and CYP3A belong to ferritin and NO forms a reversible yet stable nitrosyl complex with heme iron, which inhibits the catalytic activity of CYP3A. Additionally, CYP3A activity is inhibited by the production of peroxonitrile by NO and superoxides. Further, covalently modified P450 exists in tyrosine nitrated form, which effectively inactivates the enzyme [42, 43]. HRP inhibits the nitration of CYP3A post-translationally by inhibiting iNOS expression.

It is important to note, however, that a specific limitation was associated with this study in that rats cannot be naturally transfected with human hepatitis virus [44], and although we successfully adopted the BCG immune liver injury rat model, which is recognized by domestic and foreign scholars to simulate the immune liver injury caused by viral hepatitis, species differences and models limitations do exist. Hence, transgenic rat studies will be used for follow-up experiments to further explore the molecular mechanism. Furthermore, since the sea buckthorn granules are commonly used in Mongolian and Chinese medicine and have the same origin as various medicinal and food products, relatively few adverse reactions are associated with this treatment, which can be further confirmed by clinical experiments.

In summary, we determined that HRP inhibits the down-regulation of CYP3A during immune liver injury primarily through transcriptional regulation and post-translational nitration of enzyme proteins. This mechanism is also applicable to alcoholic liver injury and non-alcoholic fatty liver. It has been discussed in the papers we have published, and other scholars

have similar papers [4, 45]. These findings will serve to enrich the current understanding regarding the clinical effects of *H. rhamnoides* particles, while providing novel strategies for hepatitis therapeutics.

## Supporting information

**S1 Fig. Effects of HRP on NF-κB, iNOS, and CYP3A expression in rat livers following immune-mediated injury (Western blot analysis, Fig 7A).**
(PDF)

**S2 Fig. Effects of HRP on NF-κB, iNOS, and CYP3A expression in rat livers following immune-mediated injury (Western blot analysis, Fig 7B).**
(PDF)

**S3 Fig. Effects of HRP on NF-κB, iNOS, and CYP3A expression in rat livers following immune-mediated injury (Western blot analysis, Fig 7C).**
(PDF)

**S1 Data.**
(XLS)

**S2 Data.**
(XLS)

**S3 Data.**
(XLS)

**S4 Data.**
(XLS)

**S5 Data.**
(XLS)

**S6 Data.**
(XLS)

**S7 Data.**
(XLS)

**S8 Data.**
(XLS)

## Acknowledgments

The authors thank Prof. Guo-liang Zhang of the Department of Pharmacology, Basic Medical School, Beijing University, Beijing, China, for providing technical assistance.

## Author Contributions

**Conceptualization:** Yongzhi Xue.

**Data curation:** Fengting Liu.

**Formal analysis:** Tao Wang.

**Funding acquisition:** Yongzhi Xue.

**Investigation:** Tao Wang.

**Methodology:** Fengting Liu, Xiaoxia Li, Jinxue Jia, Qin Lin.

**Project administration:** Yongzhi Xue.

**Resources:** Fengting Liu, Qin Lin.

**Software:** Fengting Liu, Tao Wang, Xiaoxia Li, Jinxue Jia, Qin Lin.

**Writing – review & editing:** Yongzhi Xue.

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
