## [Decision Letter · Decision Letter 0]

11 Jun 2020

PONE-D-20-04983

Involvement of NF-κB in the reversal of CYP3A down-regulation induced by sea buckthorn in BCG-induced rats

PLOS ONE

Dear Dr. Yongzhi Xue,

Thank you for submitting your manuscript to PLOS ONE. After careful consideration, we feel that it has merit but does not fully meet PLOS ONE’s publication criteria as it currently stands. Therefore, we invite you to submit a revised version of the manuscript that addresses the points raised during the review process.

We look forward to receiving your revised manuscript.

Kind regards,

Maite Garcia Fernández-Barrena

Academic Editor

PLOS ONE

Journal Requirements:

"NO . The funders had no role in study design, data collection and analysis, decision to

publish, or preparation of the manuscript."

Reviewers' comments:

Reviewer's Responses to Questions

**Comments to the Author**

1. Is the manuscript technically sound, and do the data support the conclusions?

Reviewer #1: Yes

2. Has the statistical analysis been performed appropriately and rigorously? 

Reviewer #1: I Don't Know

3. Have the authors made all data underlying the findings in their manuscript fully available?

Reviewer #1: Yes

4. Is the manuscript presented in an intelligible fashion and written in standard English?

Reviewer #1: Yes

5. Review Comments to the Author

Reviewer #1: The manuscript is describing a mechanism by which HRP administration is able to inhibits NF-kB and reverses CYP3A down-regulation upon immunological liver damage in vivo. The data they show are clear and they described all the experimental procedures supporting the conclusion they claim.

However, there are some minor changes I would highly recommend to improve the quality of the manuscript for its publication:

1-All the symbols for statistic analysis of all the graphs from the manuscripts are very confusing and make the figures look overloaded. I would recommend to use lines to indicate the groups that are being compared and follow the commonly used symbols (*, **, ***).

2-Figure 8 is confusing because it seems that the graphs are quantifying only the images that are included in the figure, but they are calculated from three independent experiments. All of them (the three experiments) should be showed.

3-Figure 9 does not provide any added or relevant information. It is redundant with the results shown in figure 3. I would recommend to remove this figure as I think it is unnecessary to explain the conclusions or the findings achieved.

4-Instead, I would recommend to include a final figure with an schematic abstract of the mechanism described in the manuscript, that serves as support for the readers and reinforces the findings of the study.

5-Finally, I would highly recommend the authors to discuss further about the mechanism they found. I think it would be enriching for the manuscript if the authors discussed the possibility that their findings are limited to a context of immunological liver injury, or if, on the contrary, they consider that it could occur in the same way in other contexts of liver damage as alcoholic damage or dietary fat among others.

6. PLOS authors have the option to publish the peer review history of their article (what does this mean?). If published, this will include your full peer review and any attached files.

Reviewer #1: No

---

## [Author Response · Author response to Decision Letter 0]

2 Jul 2020

We have comprehensively revised the paper based on the reviewer's and editor's comments. I would like to thank the two reviewers for their full affirmation of the paper's experimental techniques, statistical methods, data usage, and standardized English expression. We have no objection to this.

Reviewer #1: The manuscript is describing a mechanism by which HRP administration is able to inhibits NF-kB and reverses CYP3A down-regulation upon immunological liver damage in vivo. The data they show are clear and they described all the experimental procedures supporting the conclusion they claim.

However, there are some minor changes I would highly recommend to improve the quality of the manuscript for its publication:

1-All the symbols for statistic analysis of all the graphs from the manuscripts are very confusing and make the figures look overloaded. I would recommend to use lines to indicate the groups that are being compared and follow the commonly used symbols (*, **, ***).

In order to improve the publication quality of the manuscript, we have revised all the figures. We have used lines to indicate the group being compared with, and followed the commonly used symbols (*, **, ***) to make the graph look more clear. .

2-Figure 8 is confusing because it seems that the graphs are quantifying only the images that are included in the figure, but they are calculated from three independent experiments. All of them (the three experiments) should be showed.

After deleting Figure 3, Figure 8 is Figure 7. The top is a gel image, and the bottom is a histogram of the number of images expressed. By convention, the gel data comes from three experiments. See the supporting data for the original image of the gel data.

3-Figure 9 does not provide any added or relevant information. It is redundant with the results shown in figure 3. I would recommend to remove this figure as I think it is unnecessary to explain the conclusions or the findings achieved.

According to the opinions of peer reviewers, Figure 3 has little meaning to the conclusions, and Figure 3 has been deleted. Figure 9 has been described in detail in the methodology and results, and the discussion part also has a large analysis of the transcription mechanism.

4-Instead, I would recommend to include a final figure with an schematic abstract of the mechanism described in the manuscript, that serves as support for the readers and reinforces the findings of the study.

According to peer reviewers, a schematic diagram of the mechanism has been added. See figure 10.

5-Finally, I would highly recommend the authors to discuss further about the mechanism they found. I think it would be enriching for the manuscript if the authors discussed the possibility that their findings are limited to a context of immunological liver injury, or if, on the contrary, they consider that it could occur in the same way in other contexts of liver damage as alcoholic damage or dietary fat among others.

This mechanism is also applicable to alcoholic liver injury and non-alcoholic fatty liver. It has been discussed in the papers we have published, and other scholars have similar papers. In order to make the paper concise, add a sentence and mark references in the conclusion: this mechanism is also applicable to alcoholic liver injury and non-alcoholic fatty liver.

 1. Lin Q, Kang X, Li X, Wang T, Liu F, Jia J, et al. NF-kappaB-mediated regulation of rat CYP2E1 by two independent signaling pathways. PloS one. 2019;14(12):e0225531. Epub 2019/12/28. doi: 10.1371/journal.pone.0225531. PubMed PMID: 31881060; PubMed Central PMCID: PMCPMC6934272.

2. Wu D, Xu C, Cederbaum A. Role of nitric oxide and nuclear factor-kappaB in the CYP2E1 potentiation of tumor necrosis factor alpha hepatotoxicity in mice. Free radical biology & medicine. 2009;46(4):480-91. Epub 2008/12/10. doi: 10.1016/j.freeradbiomed.2008.11.001. PubMed PMID: 19063961.

3. Suolang PC, Liu BQ, Chen J, De J, Nima ZB, Dunzhu CR. Protective effect and mechanism of Qiwei Tiexie capsule on 3T3-L1 adipocytes cells and rats with nonalcoholic fatty liver disease by regulating LXRα, PPARγ, and NF-κB-iNOS-NO signaling pathways. Journal of ethnopharmacology. 2019;236:316-25. Epub 2019/03/10. doi: 10.1016/j.jep.2019.03.006. PubMed PMID: 30851372.

---

## [Decision Letter · Decision Letter 1]

25 Aug 2020

Involvement of NF-κB in the reversal of CYP3A down-regulation induced by sea buckthorn in BCG-induced rats

PONE-D-20-04983R1

Dear Dr. xue,

We’re pleased to inform you that your manuscript has been judged scientifically suitable for publication and will be formally accepted for publication once it meets all outstanding technical requirements.

Kind regards,

Alfred S Lewin, Ph.D.

Section Editor

PLOS ONE

Additional Editor Comments (optional):

Reviewers' comments:

Reviewer's Responses to Questions

**Comments to the Author**

1. If the authors have adequately addressed your comments raised in a previous round of review and you feel that this manuscript is now acceptable for publication, you may indicate that here to bypass the “Comments to the Author” section, enter your conflict of interest statement in the “Confidential to Editor” section, and submit your "Accept" recommendation.

Reviewer #1: All comments have been addressed

2. Is the manuscript technically sound, and do the data support the conclusions?

Reviewer #1: Yes

3. Has the statistical analysis been performed appropriately and rigorously? 

Reviewer #1: I Don't Know

4. Have the authors made all data underlying the findings in their manuscript fully available?

Reviewer #1: Yes

5. Is the manuscript presented in an intelligible fashion and written in standard English?

Reviewer #1: Yes

6. Review Comments to the Author

Reviewer #1: In order to improve the publication quality of the manuscript I would highly recommend reviewing the aesthetics of the graphics. Many lines are not aligned as they should and the graphs look sloppy.

7. PLOS authors have the option to publish the peer review history of their article (what does this mean?). If published, this will include your full peer review and any attached files.

Reviewer #1: No

---

## [Editor Report · Acceptance letter]

31 Aug 2020

PONE-D-20-04983R1 

Involvement of NF-κB in the reversal of CYP3A down-regulation induced by sea buckthorn in BCG-induced rats 

Dear Dr. xue:

I'm pleased to inform you that your manuscript has been deemed suitable for publication in PLOS ONE. Congratulations! Your manuscript is now with our production department. 

Kind regards, 

on behalf of

Dr. Alfred S Lewin 

Section Editor

PLOS ONE